# Sex-biased gene expression is repeatedly masculinized in asexual females

Darren J. Parker[1,2]*, Jens Bast[1], Kirsten Jalvingh[1], Zoé Dumas[1], Marc Robinson-Rechavi [1,2] & Tanja Schwander[1]

Males and females feature strikingly different phenotypes, despite sharing most of their genome. A resolution of this apparent paradox is through differential gene expression, whereby genes are expressed at different levels in each sex. This resolution, however, is likely to be incomplete, leading to conflict between males and females over the optimal expression of genes. Here we test the hypothesis that gene expression in females is constrained from evolving to its optimum level due to sexually antagonistic selection on males, by examining changes in sex-biased gene expression in five obligate asexual species of stick insect, which do not produce males. We predicted that the transcriptome of asexual females would be feminized as asexual females do not experience any sexual conflict. Contrary to our prediction we find that asexual females feature masculinized gene expression, and hypothesise that this is due to shifts in female optimal gene expression levels following the suppression of sex.

[1] Department of Ecology and Evolution, University of Lausanne, 1015 Lausanne, Switzerland. [2] Swiss Institute of Bioinformatics, 1015 Lausanne, Switzerland.
*email: DarrenJames.Parker@unil.ch

Genetic constraints between developmental stages, sexes and castes arise as a result of a shared genome[1]. Species are able to mitigate these constraints by differentially expressing suites of genes in specific contexts to produce and maintain different phenotypes. This resolution, however, may be incomplete when regulatory control of gene expression is not sufficiently labile as to allow for optimal expression in each phenotypic context, leading to intralocus conflict[2].

This phenomenon has been most widely studied between the sexes, where strong sexual dimorphism is generally underlain by sex-biased gene expression[3]. Sex-biased gene expression is thought to have been largely driven by selection to resolve intralocus sexual conflict[4]. As such, contemporary sex-biased gene expression is expected to represent a combination of both resolved and partially un-resolved sexual conflict[5,6]. In the latter case, suboptimal gene expression levels are maintained by opposing selection in males and females, with the relative strengths of selection acting on each sex determining the difference between optimal and observed (suboptimal) expression levels.

Sexually antagonistic selection has the potential to constrain the optimal expression of large portions of a species' transcriptome and to thereby generate sub-optimal phenotypes in each sex. However, whether un-resolved conflict is pervasive can be difficult to investigate in natural populations, due to the relatively small (but numerous) effects of individual loci[5,7]. An ideal situation to address this question would be to examine how the transcriptome evolves following the cessation of sexual conflict. This is the case in asexually reproducing species when derived from a sexual ancestor. Because asexual species consist only of females, there is no sexual conflict and selection can optimise the female phenotype independently of any correlated effects in males. Despite the potential of this approach, previous studies have only used sexual species, examining how the transcriptome changes under experimentally altered levels of sexual selection[8–11].

The premise of these studies is that because sexual selection is typically stronger on males than females[12], a reduction in sexual selection (e.g. by enforcing monogamy) will disproportionately affect males, resulting in a shift in gene expression towards the female optimum. While this optimum is unknown, it is assumed that female-biased genes are generally beneficial for females, and male-biased genes for males[4,13], such that shifts towards female optima would generate a feminisation of gene expression (increased female-biased and decreased male-biased expression). The empirical support for this hypothesis, however, remains mixed[8–11], and the most recent study[9] further showed that shifts in sex-biased gene expression under altered sexual selection varied among tissues and conditions. However, a potential constraint in these studies is that even under reduced sexual selection, selection still acts on male phenotypes as fertile males still need to be produced in each generation. Thus, many genes potentially subject to sexually antagonistic selection therefore remain unaffected by reduced sexual selection, with genes negatively affecting male viability or fertility being obvious examples. This constraint does not apply to recently derived asexual species as all aspects of sexual conflict present in the sexual species are absent in the asexual species.

Here we use *Timema* stick insects to examine how sex-biased genes change in expression following a transition to asexuality. *Timema* comprise multiple independent transitions to asexuality (Fig. 1)[14], allowing us to examine how idiosyncratic any shifts in sex-biased gene expression are. Furthermore, male *Timema* have a single X and no Y chromosome (XX/X0 sex determination)[15], avoiding any potential difficulties arising from sex-limited regions of the genome. We first identify genes with sex-biased expression in five sexual *Timema* species by sequencing the transcriptomes of three different tissue types in each sex. We then study the fate of these sex-biased genes in close asexual relatives of each sexual species to test whether their expression is consistently feminized. This allows us to test the importance of intralocus sexual conflict on gene expression changes following a loss of sex. Contrary to our prediction, we find evidence for an overall masculinisation of sex-biased gene expression in asexual females, and hypothesise that this is due to shifts in female trait optima levels following the suppression of sex.

## Results

**Asexuality repeatedly masculinises gene expression.** To examine changes in sex-biased gene expression in asexual females, we first identified orthologous genes in each of the five sexual-asexual sister species pairs using reciprocal best Blast hits. We then classified genes as being sex-biased by comparing male and female expression in each sexual species (FDR < 0.05, absolute fold-change > 2). Sex-biased genes were identified separately for each sexual species and each of the three tissue types (whole bodies, reproductive tract and leg tissue; see Methods). As expected, given their different roles and morphology in males and females, reproductive tracts featured large numbers of sex-biased genes in all species (2843–3845, corresponding to ~30% of each transcriptome; Supplementary Table 1). Legs and whole bodies had fewer genes with sex-biased gene expression overall, but with considerable variation among species (0.5–12%; Supplementary Table 1). Variation among species could be due to variation in sexually dimorphic physiology between species but is also likely driven (at least partially) by differences in between sample variance (Supplementary Table 2). Note that because sex-biased genes were identified separately for each sexual species this approach cannot be used to determine if sex-biased genes are the same across species. We therefore investigated if sex-biased genes are the same between species as a second step (see below). Although the genes may be different, we can examine if sex-biased genes in different species are involved in similar functions by comparing the GO terms of sex-biased genes across species. Sex-biased genes in each species and tissue type were significantly enriched for many functions (136–445 significant GO terms for male-biased genes, 138–726 for female-biased genes; Supplementary Data 1). Few GO terms overlapped between species (Supplementary Fig. 1) (though the overlap was greater than expected by chance (SuperExactTest, FDR < 0.05, Supplementary Data 2)), even when enriched GO terms were first clustered by parent or child terms (Supplementary Fig. 2).

We then examined whether sex-biased genes change in expression between sexual and asexual females. Surprisingly, we found that the transcriptomes of asexual females were strongly masculinized. The expression of female-biased genes was significantly reduced in all five independently evolved asexual species and in each tissue type (14 out of 15 instances, the exception being the whole-body comparison between *T. podura* and *T. genevievae*, which shows reduced female-biased gene expression, but not significantly (Wilcoxon test, FDR = 0.076), Fig. 2, Supplementary Table 3). By contrast, male-biased genes significantly increased in expression in most tissue types of asexual females (10 out of 15 instances), although they also significantly decreased in two instances (in *T. shepardi* reproductive tracts and *T. tahoe* legs) (Fig. 2, Supplementary Table 3). We also examined if the amount of change in sex-biased gene expression altered with asexual lineage age (measured as sex-asex species divergence time[16]). While we found a relationship between sex-biased gene expression and asexual lineage age (permutation ANCOVA, $P < 0.001$), it was small and inconsistent

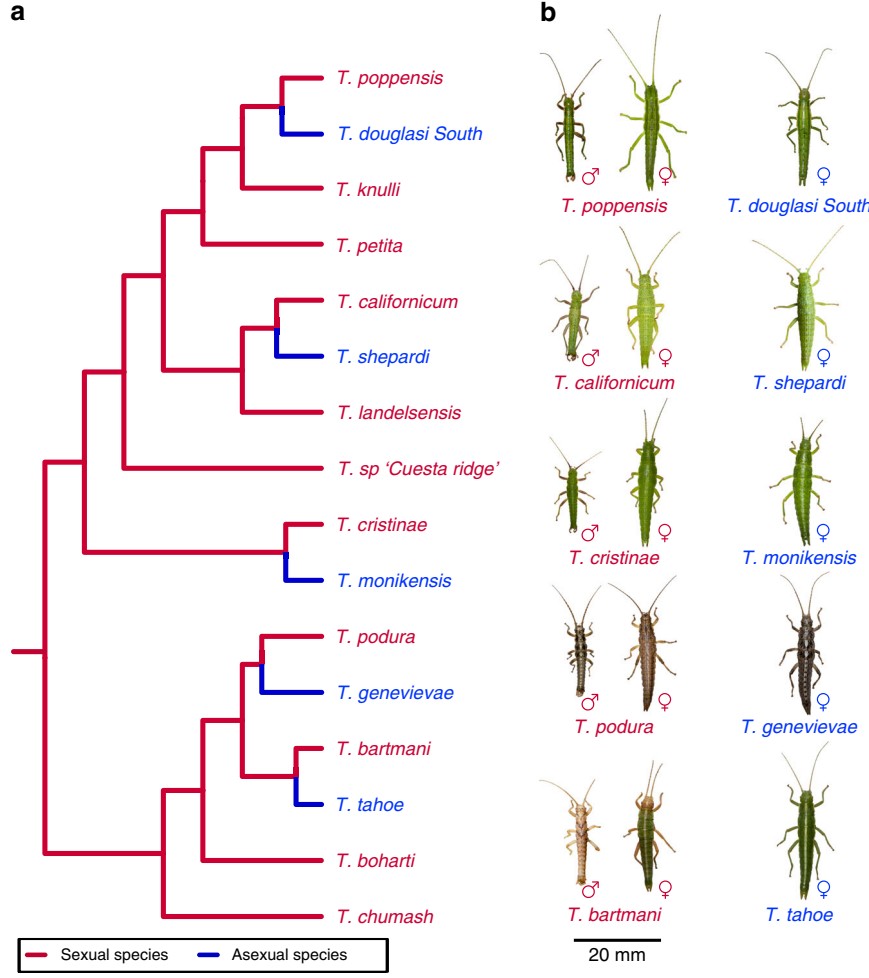

**Fig. 1** *Timema* phylogeny and photographs. **a** Phylogeny of described *Timema* species (redrawn from Riesch et al.[36]) with asexual species added from Schwander et al.[14]. Sexually reproducing species are shown in red, independently derived asexual lineages in blue. Branches between sexual-asexual sister species indicate relative divergence time based on Jukes–Cantor corrected divergence from Bast et al.[16]. Note the oldest asexual lineage, *T. genevievae*, was previously estimated to be 1.5 My old[14]. **b** Photographs of the species used in this study scaled using median body lengths from their species descriptions[42-46]. Photographs of *Timema* were kindly provided by Bart Zijlstra

between tissue-types (Supplementary Fig. 3, *P*-value of interaction term <0.001).

In addition to sex-biased genes, one class of interesting genes is sex-limited genes (genes expressed in only one of the two sexes). The expression of sex-limited genes depends on sex-specific regulation in males and females. Sex-limited genes are therefore expected to be free from sexual conflict over expression levels and may show different shifts in expression in asexual females than sex-biased genes. In particular, we expect that there will be no overall change in expression between sexual and asexual females, if relaxation of sexual conflict is the main driver of changes in asexual females. Note that sex-limited genes were identified separately from sex-biased genes to avoid inflating the dispersion of the model used to identify sex-biased genes (see Methods). Overall, we find only a few sex-limited genes (0–50), with most of these in the reproductive tracts (Supplementary Tables 4 and 5). Like female-biased genes, female-limited genes also show a significant reduction in expression in asexual females in most cases (eight out of the nine instances with more than one female-limited gene) (Fig. 3, Supplementary Fig. 4, Supplementary Table 4). Almost all male-limited genes show very little to no expression in asexual females, and are expressed at much lower levels than found in males (Fig. 3, Supplementary Fig. 4, Supplementary Table 4), suggesting that few, if any, male-

limited genes have been co-opted for new functions in asexual females.

Our analyses show that gene expression in asexual females is generally masculinized. This effect is particularly clear for female-biased genes, which decrease in expression across five different species and three different tissue types, showing the masculinisation of sex-biased gene expression in asexuals is very repeatable. Given this unexpected finding, we verified that our results were not biased by the gene sets we chose to use, which excluded genes with very low expression in asexual females, and genes without an ortholog between sexual and asexual sister species (see Methods). Exclusion of these genes could bias our results if shifts in gene expression disproportionately occur in these genes. To examine the impact of these factors we firstly repeated our analyses without excluding genes with low expression in asexual females. Generally excluded genes were few in number (1–6%) and more likely to be male-biased (Supplementary Tables 6–7). Repeating our analyses with these genes included found that shifts in sex-biased gene expression in asexuals remained qualitatively the same as when they were excluded (Supplementary Fig. 5). Secondly, we mapped reads from all samples of a sexual-asexual species pair to a single reference (the full transcriptome of either the sexual or the asexual species). With this strategy there is no need to identify orthologs between sexual and asexual species

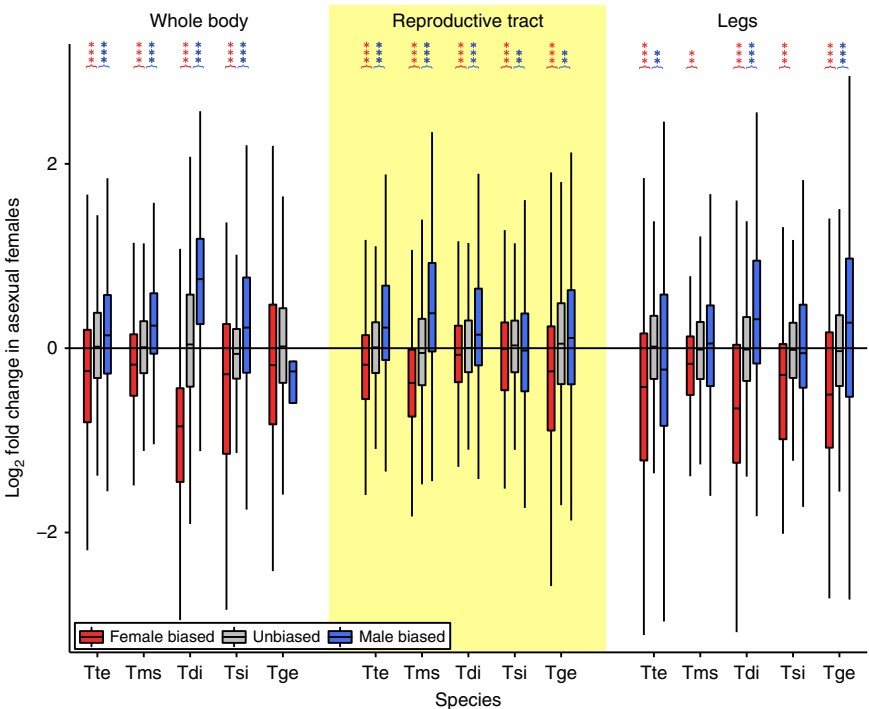

**Fig. 2** Expression shifts in sex-biased genes in asexual females. Positive values on the *y*-axis indicate increased expression in asexual females. Asterisks indicate the significance level (FDR) of Wilcoxon tests comparing the change in expression in female-biased (red) and male-biased (blue) genes to unbiased genes (\*\*\*<0.001, \*\*<0.01, \*<0.05). Species names are abbreviated as follows: Tte = *T. tahoe*, Tms = *T. monikensis*, Tdi = *T. douglasi*, Tsi = *T. shepardi*, and Tge = *T. genevievae*. Boxes represent the interquartile range (25th and 75th percentiles) of the data with the line inside the box representing the median. Whiskers show the most extreme value in the data which is no more than 1.5 times the interquartile range from the box. Source data are provided as a Source Data file

pairs. Repeating our analyses using the full sexual or asexual transcriptome, we found very few sex-biased genes had no expression in asexual females (Supplementary Tables 8–11), and we obtained qualitatively similar results as in the main analysis (Supplementary Figs. 6 and 7). Taken together these analyses show that the masculinisation of gene expression we observe is not biased by our gene set selection.

**Gene expression changes are independent of gene identity**. In the above analyses, each species-pair was treated separately, which allowed us to maximise the number of genes used in comparing changes in sex-biased gene expression between sexual and asexual females. In doing so we use five different reference gene sets (pairwise orthologs between sexual and asexual sister species, see Methods), which prevents us from examining whether repeated changes to the same sex-biased genes are responsible for the expression shifts we observe in asexual females.

To answer this question, we firstly repeated the above analyses using only genes with one-to-one orthology between all ten species (between 2886 and 3003 expressed genes depending on tissue type, see Methods). Results based on this reduced gene set are qualitatively the same as using the full gene set, i.e. an overall masculinisation of sex-biased gene expression in asexual females (Supplementary Fig. 8). As in the previous analyses the reproductive tract featured more sex-biased genes (784–1071) than whole bodies and legs (43–375) (Supplementary Table 12). This pattern is further illustrated by the fact that reproductive tract samples cluster first by sex and then phylogeny, whereas it is the opposite for legs (Fig. 4). Whole-body samples show a more mixed pattern with most samples clustering firstly by sex but with one species (*T. podura*, which has the fewest sex-biased genes in this tissue type) clustering firstly by phylogeny. Despite the lower

power of this smaller gene set (compared to the full gene set), expression of female-biased genes was significantly reduced in asexual females in 11 out of 15 instances. Male-biased gene expression significantly increased in asexual females in nine out of 15 instances (Supplementary Fig. 8, Supplementary Table 13).

The overlap between sex-biased genes from different species is significantly greater than expected by chance but rather small in size (Fig. 5a, b, Supplementary Data 3). Importantly for our analyses, the small overlap between species means that the consistently masculinized gene expression we observe in asexual females is largely independent of gene identity. This finding is strengthened by an examination of the shifts in expression for genes sex-biased in 1, 2, 3, 4 or 5 sexual species, which show that the masculinisation seen in asexual females is stronger for genes that are sex-biased in fewer sexual species (Fig. 5c, Likelihood Ratio Test, FDR < 0.05 for male and female-biased genes in all tissues, Supplementary Table 14). These findings suggest that the shifts in sex-biased gene expression we see are likely due to the property of them being sex-biased, rather than them being involved in the same specific biological process.

**Functional analysis of sex-biased genes**. A plausible explanation for decreased expression of female-biased genes in asexual females is selection against traits used for sexual reproduction. In asexual *Timema*, several sexual traits are known to be reduced, including the production of volatile and contact pheromones[17]. Here we observe that female-biased genes are indeed enriched for terms linked to the production of sexual phenotypes (e.g. pheromone biosynthetic process, reproductive behaviour, etc, Supplementary Data 1), however, to more specifically identify functions that may be affected by shifts in gene expression, we

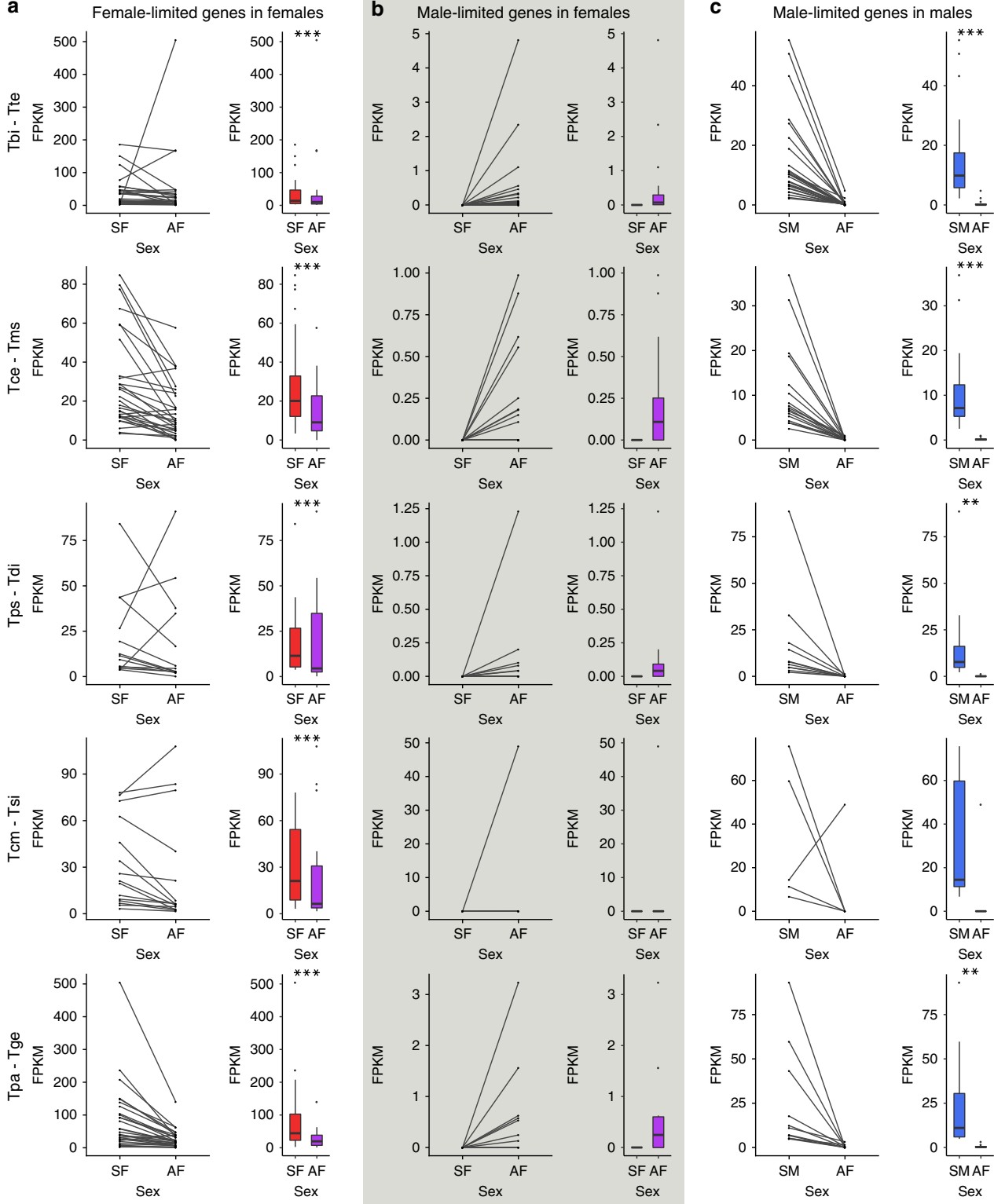

examined the GO terms specifically enriched in female-biased genes with decreased expression in asexual females.

Depending on species and tissue type, between 0 and 160 GO terms were significantly enriched, with far fewer terms enriched in legs than in whole-bodies or reproductive tracts (Supplementary Data 4), as expected given the smaller number of sex-biased genes in legs. There are no consistently enriched GO terms between all species (Supplementary Fig. 9A), and although some

terms can be easily associated with reduction of sexual traits (e.g. olfactory behaviour, chemosensory behaviour, detection of stimulus involved in sensory perception), the majority of terms have no clear link to sexual trait reduction. Most enriched terms instead are related to metabolic and developmental processes. This could potentially be a signature of a shift in energy budget in asexual females, which no longer have to produce costly sexual traits (Supplementary Data 4). Male-biased genes that increased

**Fig. 3** Expression of sex-limited genes in the reproductive tract. **a** Expression of female-limited genes in sexual females (SF, red) and asexual females (AF, purple), **b** Expression of male-limited genes in sexual females (SF, red) and asexual females (AF, purple), **c** Expression of male-limited genes in sexual males (SM, blue) and asexual females (AF, purple). Asterisks indicate the significance level (FDR) of Wilcoxon tests (***<0.001, **<0.01, *<0.05). Species names are given as abbreviations in the form sexual-species–asexual species at the left-hand side (Tbi = *T. bartmani*, Tce = *T. cristinae*, Tps = *T. poppensis*, Tcm = *T. californicum*, Tpa = *T. podura*, Tte = *T. tahoe*, Tms = *T. monikensis*, Tdi = *T. douglasi*, Tsi = *T. shepardi*, and Tge = *T. genevievae*). For the boxplots, boxes represent the interquartile range (25th and 75th percentiles) of the data with the line inside the box representing the median. Whiskers show the most extreme value in the data which is no more than 1.5 times the interquartile range from the box. Note this figure depicts only the results from the reproductive tract. For whole-bodies see Supplementary Fig. 4. Legs were not plotted due to the small number of sex-limited genes in this tissue type (Supplementary Tables 4–5). Source data are provided as a Source Data file

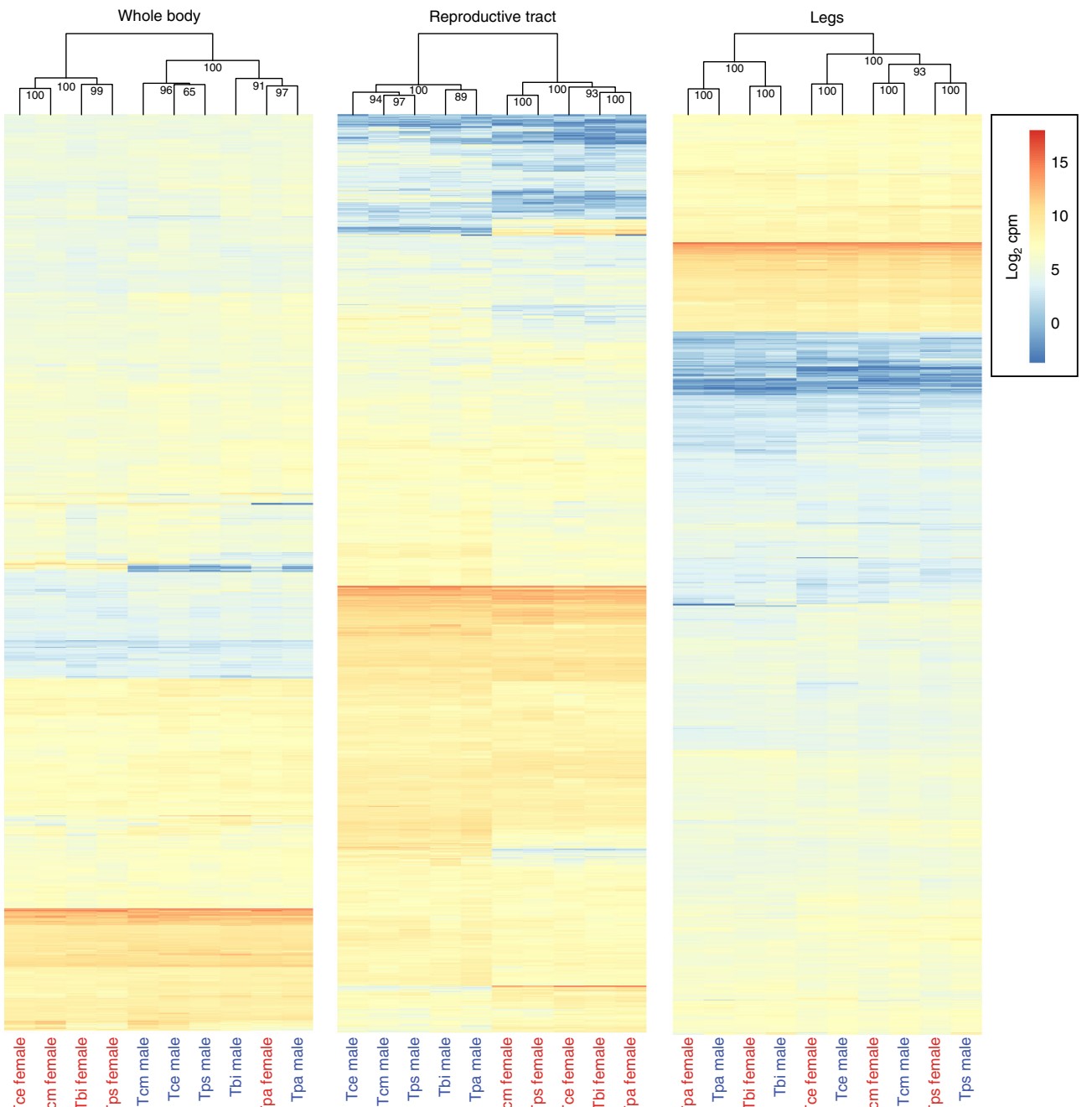

**Fig. 4** Heatmaps and hierarchical clustering of gene expression. Gene expression (log_2 CPM) for whole-body, reproductive tract and leg samples are plotted separately. Values on each node show the bootstrap support from 10,000 replicates. Species names are abbreviated as follows: Tbi = *T. bartmani*, Tce = *T. cristinae*, Tps = *T. poppensis*, Tcm = *T. californicum*, Tpa = *T. podura*. Source data are provided as a Source Data file

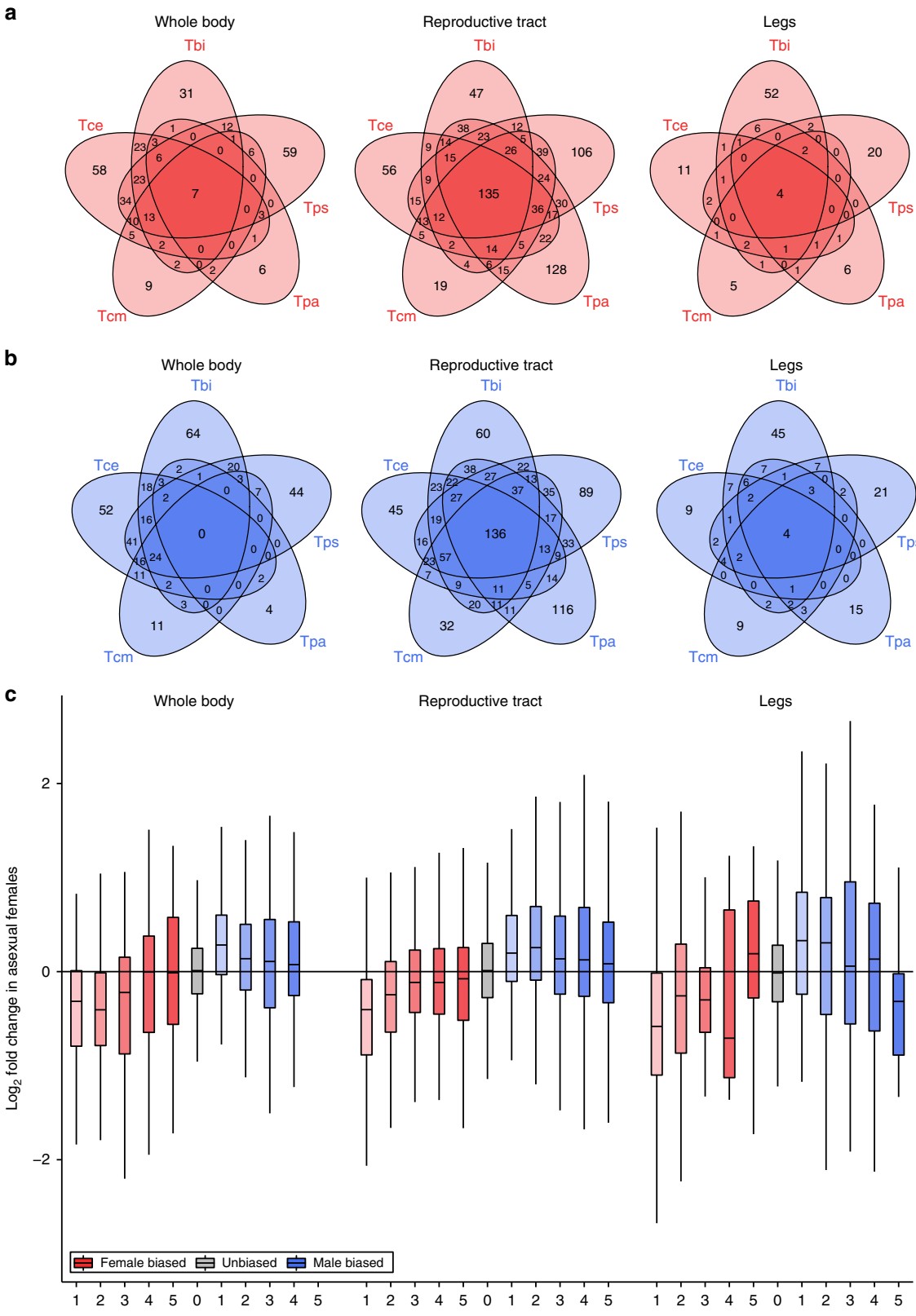

**Fig. 5** Sex-biased genes across *Timema* species. **a** Venn-diagrams showing the overlap of female-biased genes. **b** Venn-diagrams showing the overlap of male-biased genes. **c** Boxplots showing the change in expression of female-biased (reds) and male-biased (blues) genes in asexual females when a gene is female or male-biased in 1, 2, 3, 4 or 5 sexual species. Note for genes sex-biased in multiple species the plot includes fold-change values of that gene in each species it is sex-biased in. Boxes represent the interquartile range (25th and 75th percentiles) of the data with the line inside the box representing the median. Whiskers show the most extreme value in the data which is no more than 1.5 times the interquartile range from the box. Source data are provided as a Source Data file

in expression were enriched for between 0 and 81 terms, and again, no terms were shared between all species, and very few between any pair of species (Supplementary Data 5, Supplementary Fig. 9B).

The removal of sexual conflict is expected to cause the feminisation of gene expression in asexual females. Although overall the pattern of expression change we observe is opposite to this prediction, it is possible that a feminisation of gene expression still occurs for a small subset of genes, but its effect is masked by the larger effect of masculinisation. We specifically examine the subset of sex-biased genes that follow the expected pattern of feminisation, by looking at processes enriched for female-biased genes that increase in expression and male-biased genes that decrease in expression in asexual females. We would expect that genes showing feminisation would be enriched for processes associated with sexual conflict. Both male- and female-biased genes showed an enrichment of many terms (between 0 and 360, and between 1 and 195, respectively), including some that could be associated with sexual conflict (e.g. sexual reproduction, female mating behaviour, etc). However, the majority of terms have no clear link to sexual conflict, and again no terms were shared between all species (Supplementary Data 6, 7, Supplementary Fig. 10).

Taken together, the functional enrichment analyses suggest that the changes in gene expression we observe are involved in a diverse set of processes in each of the species. This is in line with what we observe from the gene expression analyses, which show that sex-biased genes have little overlap between species, and that the largest shifts in gene expression are in genes that are sex-biased in the fewest species. In addition to being different between species, the enriched GO terms were not particularly informative for determining if they are involved in sexual trait decay or sexual antagonism. This reflects the relative difficulty in obtaining functional annotations in *Timema*, due to their evolutionary distance from a well characterised insect model system, meaning that most functions are broad and difficult to attribute to specific roles in sexual traits or sexual antagonism.

**Sequence divergence of sex-biased genes in asexuals.** Sex-biased genes in sexual species often evolve rapidly, because of relaxed evolutionary constraint[3] or due to strong sexual selection and/or sexual antagonism which drives positive selection for amino-acid changes[18]. In asexual species, sex-biased genes are also expected to evolve rapidly, but due to reduced purifying selection on redundant sexual traits underlain by sex-biased genes. Although interesting, identifying differences in evolutionary rates between gene classes in asexual species is difficult due to the overall elevated rates in asexual species (including in *Timema*[16]), and because genes are inherited as a single linkage group, which reduces the power to detect differences in evolutionary rate between genes. Here we found evidence for an elevated rate of dN/dS in asexual species and in sex-biased genes (Supplementary Fig. 11, Supplementary Tables 15–17). We do not see any evidence for an interaction between sex-bias and reproductive mode (Supplementary Fig. 11, Supplementary Tables 15–17), indicating that the increase in dN/dS for sex-biased genes is similar in sexual and asexual species.

## Discussion

Conflict over gene expression levels between males and females is thought to drive the evolution of sex-biased gene expression[4]. While sex-biased expression is expected to reduce the amount of intralocus sexual conflict, it is unlikely to be complete for many genes, meaning that some proportion of sex-biased genes are likely subject to sexually antagonistic selection[5,6]. Here we chose

to investigate how sex-biased gene expression changes in asexual species which experience no sexual conflict. We predicted that transcriptomes of asexual females would be feminized as sex-biased genes in asexual females would no longer be constrained by countervailing selection pressures in males. Contrary to our prediction we found evidence for an overall masculinisation of sex-biased gene expression in asexual females. This pattern of masculinisation was very consistent across each of the five independently derived asexual species, and three tissue types we examined. In addition, masculinisation was not driven by changes in expression of the same genes in each species, showing that it is the property of being sex-biased per se that is most likely to be responsible for the shifts in expression we observe.

Taken together, our results provide strong evidence for a masculinisation of gene expression in asexual species. The strength of this finding does not mean there is no sexual conflict over optimal levels in sexual species, but rather that changes in asexual females driven by a release of conflict are negligible relative to changes driven by other mechanisms. The presence of such alternative mechanisms can best be illustrated by the fact that female-limited genes (that should experience no sexual conflict over gene expression level in sexual species), show a consistent masculinisation similar to sex-biased genes. We suggest that this is because, although reproducing asexually does remove the pressure of sexual conflict, it also removes the need for many of the sexual traits sexually dimorphic gene expression underlies. Consequently, while we expected gene expression in asexual females to be free to move to a female optimum, it is also likely that the optimal female phenotype is different for sexual and asexual females.

Female asexual *Timema* show reductions in several sexual traits including a reduced sperm storage organ, and reduced volatile and contact pheromone production[17]. Since sexually dimorphic traits are largely a product of sex-biased gene expression[3], a link between reduced female sexual traits and reduced female-biased gene expression is a plausible explanation for the decreased expression of female-biased genes we observe. It is less clear why we also see an accompanying increase of expression in male-biased genes in asexual females. We suggest four, non-mutually exclusive, speculative explanations for this. Firstly, increased expression of male-biased genes may arise as a result of sexual trait reduction in cases where high expression of a gene in males acts to suppress the development of a trait, or when low expression in females acts to enhance a female sexual trait. In such genes selection for sexual trait reduction in asexual females would be expected to produce an increase in expression. A second potential explanation is that in sexual species there are a number of products produced by males and then transferred to females that are important for female fertility. For instance, in many insects, ovulation and oviposition are stimulated by substances in the male ejaculate such as juvenile hormone, prostaglandins, and myotropins[19,20]. Since these products are not provided by males in asexual species, females may need to increase expression of the genes that produce these products to compensate. While this explanation could explain some of the increased expression of male-biased genes we observe in the reproductive tract, it is unlikely to be a general explanation for the increased expression of male-biased genes across all species and tissue types. A third potential explanation is that if males and females in sexual species have separate niches, a transition to asexuality would allow asexual females to expand into the male niche. Differential niche use is likely to, at least in part, be mediated by sex-biased gene expression, meaning that asexual females would need to masculinise their gene expression in order to occupy the vacant niche left by males. While differences in male and female niche use have not been extensively studied in *Timema*, sexual dimorphism in

mandible shape has been reported, implying that there may be some differential use of niche-space in sexual *Timema* species[21]. Future work examining sexual niche usage and gene expression is needed to evaluate this hypothesis. Finally, another potential explanation for masculinisation of gene expression in asexual *Timema* is the decay of dosage compensation. *Timema* have an XX/X0 sex-determination system[15], meaning that in sexual species the X chromosome is present as a single copy in males and as two copies in females. *Timema* are likely to have evolved dosage compensation to equalise expression of X-linked genes between the sexes. In asexual species selection for dosage compensation is absent, which could lead to expression changes of X-linked genes. Changes in X-linked genes alone are, however, unlikely to explain the masculinized gene expression of asexual females as masculinisation is also observed in the asexual species *T. monikensis* when only autosomal transcripts are considered (Supplementary Fig. 12). While we cannot formally distinguish between X-linked and autosomal transcripts in the other asexual species because the relevant genomes are not available, these species would most likely also feature masculinisation for autosomal transcripts. Indeed, different sexual *Timema* species are characterised by quite different sets of sex-biased genes (Fig. 5) yet there is no evidence of X-chromosome turnover in *Timema*[15].

For the reasons detailed above, we believe that female trait reduction is the most likely explanation for the majority of changes in sex-biased gene expression we observe, rather than the cessation of sexual conflict. Similar to our findings, a recent study[9] found that experimentally reduced sexual selection also produced an overall masculinisation of gene expression in *D. pseudoobscura*. However, Veltsos et al.[9] interpret their findings as a consequence of reduced conflict, and attribute masculinisation to the dynamic nature of sexually antagonistic selection causing unpredictable changes in sex-biased gene expression. An alternative explanation, however, is that the masculinisation of gene expression in females seen in Veltsos et al. corresponds to a reduction of sexual traits under reduced sexual selection, similar to our findings in *Timema*. Previous studies have reported reduced sexual traits in females evolving under reduced sexual selection in the *D. pseudoobscura* lines studied by Veltsos et al.[11,22]. As such, both Veltsos et al. and our results highlight that the shifts in sex-biased gene expression we observe in the absence, or under reduced levels, of sexual conflict may be in part due to a shift in the optimal trait levels in females. Such shifts in female optima under different sexual selection scenarios are important to consider as an explanation even for studies that observe the expected feminisation of sex-biased gene expression[8,10,11]. This is because reducing sexual selection can also favour the increased expression of female sexual traits under some conditions. In these situations, the feminisation of sex-biased gene expression can be due to changes in sexual trait optima rather than due to a reduction in the amount of intralocus sexual conflict. More generally, optimal values for traits should be affected by the nature and level of sexual conflict present in a population. Changes to optimal trait values under different selective scenarios are however difficult to predict a priori[23], meaning future studies will require careful examination of optimal phenotypes under different selective scenarios in order to correctly interpret any changes in sex-biased gene expression.

In conclusion, we find that sex-biased gene expression is repeatedly masculinized following a transition to asexuality, and suggest that this result is driven primarily by a reduction of female sexual traits. While we observe similar patterns of masculinisation across all five asexual species, the genes involved were mostly different, reflecting the dynamic nature of sex-biased gene expression. In line with this, the functional processes associated with expression change in each species were also diverse. Finally,

our study highlights the importance of considering explanations other than intralocus sexual conflict for explaining shifts in sex-biased gene expression, since differences in sexual conflict are also likely to be accompanied by changes in sexual trait optima, which may enhance or mask changes caused by a reduction or cessation of intralocus sexual conflict.

## Methods

**Samples**. Individuals for whole-body and tissue-specific samples were collected from the field as last instar juveniles in spring 2013 and 2014, respectively (collection locations for all samples are given in Supplementary Data 8). All individuals were raised in common garden conditions (23 °C, 12 h:12 h, 60% humidity, fed with Ceanothus cuttings) until 8 days following their final moult. Prior to RNA extraction, individuals were fed with an artificial medium for 2 days to avoid RNA contamination with gut content and then frozen at −80 °C. For leg samples, three legs were used from each individual (one foreleg, one midleg and one hindleg). Reproductive tracts were dissected to consist of ovaries, oviducts and spermatheca in females and testes and accessory glands in males. Note the same individuals were used for leg and reproductive tract samples. To ensure individuals were reproductively active at the time of sampling, all sexual individuals were allowed to mate, and asexual and sexual females were observed to lay eggs. When analyses were repeated using virgin sexual females, we obtained qualitatively similar results (Supplementary Fig. 13). Note only whole-body samples were available for this comparison. Ethical approvals or collection permits were not required for this research.

**RNA extraction and sequencing**. We generated three biological replicates per species and tissue type from pooled individuals (1–9 individuals per replicate, a total of 516 individuals, in 150 replicates in total (including the virgin sexual females); see Supplementary Data 8). To extract RNA, samples were flash-frozen in liquid nitrogen followed by addition of Trizol (Life Technologies) before being homogenised using mechanical beads (Sigmund Lindner). Chloroform and ethanol were then added to the samples and the aqueous layer transferred to RNeasy MinElute Columns (Qiagen). RNA extraction was then completed using an RNeasy Mini Kit following the manufacturer's instructions. RNA quantity and quality was measured using NanoDrop (Thermo Scientific) and Bioanalyzer (Agilent). Strand-specific library preparation and single-end sequencing (100 bp, HiSeq2000) were performed at the Lausanne Genomic Technologies Facility.

The 150 libraries produced a total of just under 5 billion single-end reads. Six whole-body and six tissue-specific libraries produced significantly more reads than the average for the other samples. To reduce any influence of this on downstream analyses, these libraries were sampled down to approximately the average number of reads for whole-body or tissue-specific libraries respectively using seqtk (https://github.com/lh3/seqtk Version: 1.2-r94).

**Transcriptome references**. De novo reference transcriptome assemblies for each species were generated previously[16]. Our expression analyses were conducted using two sets of orthologs. Firstly, we identified orthologs between sexual and asexual sister species using reciprocal Blast as described in Parker et al.[24]. Secondly, we used the 3010 one-to-one orthologs present in all 10 *Timema* species as identified by Bast et al.[16]. The identified ortholog sequences varied in length among different species. Since length variation might influence estimates of gene expression, we aligned orthologous sequences using PRANK (v.100802, default options)[25] and trimmed them using alignment_trimmer.py[26] to remove overhanging gaps at the ends of the alignments. If an alignment contained a gap of greater than three bases then sequence preceding or following the alignment gap (whichever was shortest) was discarded. Any orthologous sequences that had a trimmed length of <300 bp were also discarded. Finally, before mapping, genes with significant Blast hits to rRNA sequences were removed from the trimmed transcriptome references.

**Read trimming and mapping**. Before mapping, adapter sequences were trimmed from raw reads with CutAdapt[27]. Reads were then quality trimmed using Trimmomatic v 0.36[28], clipping leading or trailing bases with a phred score of <10 from the read, before using a sliding window from the 5′ end to clip the read if 4 consecutive bases had an average phred score of <20. Any reads with a sequence length of <80 after trimming were discarded. Reads from each libret were then mapped to the transcriptome references using Kallisto (v. 0.43.1)[29] with the following options -l 210 -s 25–bias–rf-stranded for whole-body samples and -l 370 -s 25–bias–rf-stranded for tissue-specific samples (the -l option was different for whole-body and tissue-specific samples as the fragment length for these libraries was different).

**Differential expression analysis**. Expression analyses were performed using the Bioconductor package EdgeR (v. 3.18.1)[30] in R (v. 3.4.1)[31]. Firstly, to identify sex-biased genes we compared male and female expression separately for each tissue type in each sexual species. Genes with counts per million <0.5 in 2 or more libraries per sex were excluded from expression analyses. Normalisation factors for

each library were computed using the TMM method. To estimate dispersion, we then fit a generalised linear model (GLM) with a negative binomial distribution with sex as an explanatory variable and used a GLM likelihood ratio test to determine the significance of sex on gene expression for each gene. P-values were then corrected for multiple tests using Benjamini and Hochberg's algorithm[32]. Sex-biased genes were then defined as genes that showed a greater than 2 fold difference in expression between males and females with an FDR < 0.05. Note all genes not classified as sex-biased were classified as unbiased genes. We chose this threshold in order to select a robust set of sex-biased genes, and to reduce the effect of sex-biased allometry[33]. Note that analyses using just an FDR threshold to define sex-biased genes, or using virgin sexual females to independently verify sex-biased genes in whole-body samples, produced qualitatively similar results (Supplementary Tables 18–19, Supplementary Fig. 14).

Clustering of expression values (log$_2$ CPM) was performed using Ward's hierarchical clustering of Euclidean distances with the R package pvclust (v. 2.0.0)[34], with bootstrap resampling (method.hclust = "ward.D2", method.dist = "euclidean", nboot = 10000), and visualised using R package pheatmap (v. 1.0.8)[35].

To quantify how sex-biased genes change in expression in asexual females we then compared gene expression in sexual and asexual females separately for each species pair and each tissue type. We also compared the change in expression in asexual females for male- and female-biased genes to unbiased genes using a Wilcoxon test, corrected for multiple tests using Benjamini and Hochberg's algorithm[32]. To determine if changes in sex-biased gene expression in asexual females are larger for genes sex-biased in fewer species we fit a generalised linear mixed model with the number of species a gene is sex-biased in as a fixed effect and gene ID as a random effect in R. The significance of terms was determined using a Likelihood Ratio Test. A separate model was fit for male- and female- biased genes in each tissue. P-values were corrected for multiple tests using Benjamini and Hochberg's algorithm. We also examined gene expression changes in the *T. cristinae–T. monikensis* species pair when X-linked transcripts were excluded. X-linked transcripts were determined in these species by blasting (blastN) transcripts to the *T. cristinae* reference genome, for which linkage groups have been assigned[36]. The gene expression analyses were then repeated on only those transcripts that had a significant blast hit (e-value < 1 × 10$^{-20}$, query coverage > 60%) to a scaffold in an autosomal linkage group.

**Shifts in sex-biased genes and asexual lineage age.** The asexual species differ in age as estimated previously[2]. Since the age of asexuality varies we tested if changes in sex-biased gene expression altered with asexual species age using a permutation ANCOVA (number of permutations = 10,000) separately for male- and female-biased genes with the following terms: asexual lineage age, tissue type and their interaction.

**Analysis of sex-limited genes.** Sex-limited genes were classified as genes that had at least two Fragments Per Kilobase Million (FKPM) in each replicate of one sex and 0 FKPM in each replicate of the other sex. FKPM values were calculated using EdgeR. The expression levels of female-limited genes in sexual and asexual females, and male-limited genes in sexual males and asexual females were compared using a Wilcoxon test, corrected for multiple tests using Benjamini and Hochberg's algorithm[32].

**Sequence evolution of sex-biased genes.** To test if sex-biased genes have a higher rate of divergence in asexuals, we examined if sex-biased genes have elevated dN/dS ratios in asexuals. To do this we firstly fit a binomial glmm (dN/dS values were transformed to fall into two categories: zero or non-zero), with reproductive mode, sex-bias and their interaction as fixed effects and gene identity as a random effect. Secondly we firstly fit a glmm with a gamma distribution to the dN/dS values that were greater than zero, with the same fixed and random effects as the binomial model. All glmms were fit using the lme4 package (v. 1.1.14)[37] in R, and significance of terms was determined using a log-likelihood ratio test. dN/dS values were calculated for each of the one-to-one orthologs using codeml implemented in the PAML package[38] to generate maximum likelihood estimates of dN/dS for each terminal branch in the phylogeny (using the "free model") as described in Bast et al.[16].

**GO term analysis.** Genes were functionally annotated using Blast2GO (version 4.1.9)[39] as described in Parker et al.[24]. Briefly, sequences from each sexual species were compared with BlastX to either NCBI's nr-arthropod or *Drosophila melanogaster* (drosoph) databases, to produce two sets of functional annotations, one derived from all arthropods and one specifically from *Drosophila melanogaster*. The *D. melanogaster* GO term annotation generated around four times more annotations per sequence than NCBI's nr-arthropod database. We therefore conducted all subsequent analyses using the GO terms derived from *D. melanogaster*, but note that results using the annotations from all arthropods were qualitatively the same (see Supplementary Fig. 15).

To identify overrepresented GO terms we conducted gene set enrichment analyses (GSEA) using the R package TopGO (v. 2.28.0)[40], using the elim algorithm to account for the GO topology. GO terms were considered to be significantly enriched when p < 0.05.

Since we defined sex-biased genes with both FDR and FC thresholds, we ranked sex-biased genes for the GSEA to take both FDR and FC into account. To identify

overrepresented GO terms for female-biased genes, genes were ranked by FDR in four subsets: female-biased with FC > 2, female-biased with FC < 2, male-biased with FC < 2, and male-biased with FC > 2. Female-biased gene subsets were ranked so that small FDR values were ranked highly, male-biased gene subsets were ranked so that small FDR values were ranked low in the list. The four lists were then joined together in the order given above, and assigned a unique rank. This ranked list produces a list where strongly female-biased genes are at the top, followed by weakly female-biased genes, then weakly male-biased genes, and finally strongly male-biased genes at the bottom. To identify overrepresented GO terms for male-biased genes the ranked list for female-biased genes was simply inverted. Finally, to examine the GO terms overrepresented in sex-biased genes which changed expression in asexuals, female- and male-biased genes were ranked by fold-change between sexual and asexual females.

To determine if the overlap of sets of sex-biased genes or GO terms was greater than expected by chance we used the SuperExactTest package (v. 0.99.4;[41]) in R, which calculates the probability of multi-set intersections. P-values were multiple test corrected using Benjamini and Hochberg's algorithm implemented in R.

## Data availability

Raw reads have been deposited in SRA under accession codes SRR5748941-SRR5749000 for whole-body samples and SRR5786827-SRR5786961 for tissue-specific samples. The transcriptome assemblies used in this project are available from DDBJ/EMBL/GenBank under the BioProject PRJNA380865 with the following accession codes: GFPP00000000, GFPR00000000, GFPS00000000, GFPT00000000, GFPU00000000, GFPV00000000, GFPW00000000, GFPX00000000, GFPY00000000 and GFPZ00000000. The genome and linkage map used for *T. cristinae* is available from NCBI accession number: GCA_002928295.1. dN/dS values are archived at Zenodo: https://doi.org/10.5281/zenodo.3451445. The source data underlying Figs. 2–5 and Supplementary Figs. 3–8 and 11–14 are provided as a Source Data file.

## Code availability

Scripts for the analyses in this paper are available at https://github.com/DarrenJParker/Timema_Sex_Biased_Gene_Exp, and are archived at Zenodo at https://doi.org/10.5281/zenodo.3451445.

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

## Acknowledgements

This study was supported by Swiss FNS grants PP00P3_170627, PP00P3_139013 and CRSII3_160723. We would like to thank Chloe Larose and Bart Zijlstra for their assistance in the field. We also thank Jarrod Hadfeild for statistical advice.

## Author contributions

T.S. and D.J.P. designed the study. Z.D., K.J., J.B. and T.S. collected samples, performed dissections and molecular work. D.J.P. analysed the data with input from J.B., T.S. and M.R.R. D.J.P. and T.S. wrote the paper with input from all authors.

## Competing interests

The authors declare no competing interests.
