## [Peer Review File · Nature Communications]

Reviewers' Comments:

Reviewer #1:

Remarks to the Author:

This manuscript describes a study of sex-biased gene expression associated with transitions to asexuality. The study takes advantage of a remarkable system, *Timema* walking sticks, where there have been multiple transitions to asexuality. The authors compare gene expression in males and females of 5 sexual species and 5 asexual relatives, in whole bodies and 2 tissues.

This is a really interesting, timely, novel and well-conducted study. It tackles an important area of interest in the field: to what extent does the shared genome impose a constraint on the evolution of sexual dimorphism? The question is mostly studied in sexual species. This study presents a fresh look at the question and addresses a robust hypothesis: that if intralocus conflict is an important constraint, then the genomes of asexual females, freed from that constraint, should evolve more feminization.

The *Timema* system offers a unique and powerful resource with which to test the hypothesis. The study is well designed, with 5 independent transitions to asexuality, two tissue types plus whole body samples, and reasonable biological replication. The manuscript is well-presented, with clear figures.

That the result turned out to be the opposite is surprising. The authors have been careful in exploring the result. Strengths of the approach and analyses include the comparison of 2 sets of orthologs; defining sex-biased expression by both fold-change threshold and p-value (rather than relying on p-value alone), but also exploring whether other approaches change the results; and exploring how genes that were lowly-expressed in asexuals might have influenced the results. The authors judiciously weigh alternative explanations for the result.

I have a number of questions and comments.

1. Line 453-454: The authors looked at dN/dS ratios to test whether sex-biased genes are under reduced purifying selection. But although dN/dS tests can tell us about rates of evolution, they aren't useful for distinguishing whether e.g. faster rates happen through increased positive selection or relaxed purifying selection. For this purpose the authors could use the direction of selection test (Stoletzki and Eyre-Walker 2011 *Mol Biol Evol* 28: 63-70) or related tests (e.g. Eyre-Walker and Keightley 2009 *Mol Biol Evol* 26: 2097-2108). Or if polymorphism data aren't available for these tests, simply describe the dN/dS results with reference to rates of divergence rather than purifying selection.

2. Genes on autosomes versus sex chromosomes show different patterns of sex-biased expression and different rates of evolution, and dosage compensation can contribute to sex-biased expression on the X or obscure it. What happens to the X with the transition to asexuality? E.g. one possible scenario is that sexual species have one copy of the X turned off, and asexual species transition to expressing both copies; if male-biased genes are over-represented on the X, as they are in some species, then male-biased genes will be expressed more in asexuals as a consequence of the relaxed suppression of one X.

Are genes mapped such that the authors can test for different patterns between autosomes and X? If not perhaps the authors could discuss whether sex chromosome dynamics might have contributed to the observed patterns.

3. It would be more precise to say that the genome of asexuals is defeminized, and to a lesser extent, masculinized. There's a more consistent decrease in asexuals in the bias of female-biased genes than

there is an increase in the bias of male-biased genes.

4. How were unbiased genes defined? Are they any genes that weren't male- or female-biased? Defining them this way is likely to include many genes that are actually sex-biased but where the magnitude is small or variable enough that they don't pass the FDR threshold. Female-biased genes are likely to be over-represented in an unbiased gene set defined in this way, as they often have lower magnitude of bias.

5. What happens with body size with the transition to asexuality? Is there sexual size dimorphism in sexual species, and do asexual females change towards an intermediate size, as might be expected if they are covering more niche space that was split between the sexes in sexual species? If this is the case then many of the metabolic and developmental process GO terms might relate to changes in growth and body size.

6. Line 108: what level of GO term? One can imagine that more fine-grained terms might vary a great deal but higher order terms might be shared across gene sets.

7. Minor comments and typos

Headings in figure 2 – B and C are both labelled 'Male-limited' and it might help the reader grasp the meaning more quickly if they are given different headings

Line 536 typo 'od'

Line 546 genes should be gene

The Methods describes the tissues as 'tissue type', which seems appropriate as whole body is not really a tissue. The rest of the MS refers to tissues, though. It would be clearer to stick to 'tissue type' throughout the MS.

Line 17: Alternative splicing is another major form of resolution, so one might phrase this as e.g. that the resolution includes genes being expressed at a different level.

Line 247-248: In some systems there's support for faster evolution of sex-biased genes through positive selection, but in other systems, there's support for rapid evolution through relaxed constraint (Mank 2017 Nature Ecology and Evolution, The transcriptional architecture of phenotypic dimorphism)

Line 365-366: I think it is worth stating in the main text that there are only whole body samples for virgin females, to save the reader from having to refer to the supplemental materials for this.

Reviewer #2:

Remarks to the Author:

I very much enjoyed reading the manuscript by Parker et al and think it makes a very important contribution to the literature. They contrast gene expression between asexual/sexual pairs of *Timema* stick insects with the aim of testing whether females are constrained from evolving towards their transcriptional optima by sexual conflict. Instead, they find that gene expression is masculinized and attribute this to a change in fitness optima in asexual females. The approach is novel, the manuscript is very thorough and extremely well written. It deals with a hot topic in evolutionary biology at the moment, to what extent does sexual conflict act as a constraint and what is the role of gene expression in its resolution, and will therefore appeal to a wide readership. I particularly like the new perspective they provide on focusing on shifts in female fitness optima. I have a number of comments.

The main focus of the manuscript is on sex-biased genes. The authors do mention that while differential gene expression can resolve conflict, often the conflict is only partially resolved. Therefore, it does make sense to study shifts in expression of sex-biased genes when sexual conflict is

eliminated. However, I would have thought that a more interesting group of genes are those with unbiased expression. While many unbiased genes might not be subject to conflict, theory predicts that many are under strong sexual antagonism. But due to pleiotropy etc this conflict cannot be resolved by differential gene expression evolution. Unbiased genes are therefore 'hotspots' of unresolved sexual conflict. I wonder instead whether the authors could focus on these genes and identify whether there are any convergent shifts in expression in across asexual/sexual pairs? We might expect to see a subset of unbiased genes with consistently higher expression in asexual females. From Fig 1. there does appear to be shifts in expression for unbiased genes, and it would be interesting to see if the same sets of genes show similar patterns across pairs. I wonder if this is more of a direct test of the effect of eliminating sexual conflict and would also provide insight into the types of genes subject to the greatest conflicts in sexual species.

The manuscript would benefit from a figure explicitly showing the phylogenetic relationships between the five pairs of asexual/sexual species. Are the pairs equally distantly related? Variation in divergence time could help interpret why the magnitude of expression shifts differ across pairs of species. Did the authors look into this? This could provide insight into how quickly expression shifts can evolve.

A related point is whether the extent of sexual conflict in the sexual species is thought to be similar? Do the sexual species have similar mating systems and male promiscuity?

It's great to see the authors discuss confounding effects of allometry for gene expression analyses. Do they know whether there might be morphological differences across asexual/sexual pairs? Does relative ovary size differ? Are there differences in egg production? I am confident that the stringent fold change threshold they have imposed has likely eliminated any problems due to allometry. I also think they can use Fig 1 to reinforce this. Median expression of unbiased genes appears not to have changed between sexual and asexual females in each pair. If there were substantial allometry differences I think they would detect shifts in the median expression of unbiased genes, which they don't. It would be worth mentioning this in the manuscript to reinforce that allometry isn't a problem here.

I get that the authors are trying to illustrate that shifts in expression are not specifically linked to gene function. But unfortunately, I am not overly convinced by Fig 4. Fig 4A and B don't present anything new in the literature, I think it is quite well established that sex-biased genes change quickly across species. Also, the trend that masculinization is stronger for genes that are sex-biased in sexual species seems quite noisy in Fig 4C. For example, in the legs this pattern is driven by only one of the five data points. Instead, the authors could make this more convincing by conducting a statistical test to assess this relationship. I am also not clear how the y axis is calculated – is this an average across all asexual females?

There is quite a strong emphasis on gene ontology analyses in the paper. The authors themselves discuss the limitations of this approach in *Timema* due to unreliable gene annotation models. The analyses are also quite descriptive. This is the weakest part of the paper and detracts from the rest of the manuscript. I would scale down this discussion (or move it to the SI) and use the space to explore the analyses I recommend above.

The authors mention that they examine three different tissues types but really these are not independent. Whole body is an amalgamation of reproductive tract and legs. I am not sure what the whole-body results add to the message and would move this to the supplementary material.

The species names used to refer to each asexual/sexual pair vary across figures, tables and the manuscript (e.g. the species names in Fig 1 differ to ST5). This is a bit confusing. A figure of the

phylogeny would help with this but I would recommend referring to the same focal species consistently.

L20 Missing 'in'

L124 I don't quite follow this sentence. There is no Y chromosome so why do the authors think that sex-limited genes have different genetic architecture than sex-biased genes?

L149 Typo – repeated were

L447 Typo – was should be were

L463 Typo – remove .

L468 Typo – remove .

L489 Typo – remove with

L498 Typo – remove for

The formatting of the supplementary tables varies. Sometimes column names are in bold and in others not. Some have shaded boxes and others not.

Fig 1 Legend L541 Typo - gene

Fig 2 Legend L531 Typo – of

Fig 2 Legend L531 ST 6 & 7 do not list the number of sex-limited genes and so I am not sure it is relevant to refer to them here.

Reviewer #1 (Remarks to the Author):

This manuscript describes a study of sex-biased gene expression associated with transitions to
asexuality. The study takes advantage of a remarkable system, *Timema* walking sticks, where there
have been multiple transitions to asexuality. The authors compare gene expression in males and
females of 5 sexual species and 5 asexual relatives, in whole bodies and 2 tissues.

This is a really interesting, timely, novel and well-conducted study. It tackles an important area of
interest in the field: to what extent does the shared genome impose a constraint on the evolution of
sexual dimorphism? The question is mostly studied in sexual species. This study presents a fresh look
at the question and addresses a robust hypothesis: that if intralocus conflict is an important
constraint, then the genomes of asexual females, freed from that constraint, should evolve more
feminization.

The *Timema* system offers a unique and powerful resource with which to test the hypothesis. The
study is well designed, with 5 independent transitions to asexuality, two tissue types plus whole
body samples, and reasonable biological replication. The manuscript is well-presented, with clear
figures.

That the result turned out to be the opposite is surprising. The authors have been careful in
exploring the result. Strengths of the approach and analyses include the comparison of 2 sets of
orthologs; defining sex-biased expression by both fold-change threshold and p-value (rather than
relying on p-value alone), but also exploring whether other approaches change the results; and
exploring how genes that were lowly-expressed in asexuals might have influenced the results. The
authors judiciously weigh alternative explanations for the result.

I have a number of questions and comments.

1. Line 453-454: The authors looked at dN/dS ratios to test whether sex-biased genes are
under reduced purifying selection. But although dN/dS tests can tell us about rates of evolution,
they aren't useful for distinguishing whether e.g. faster rates happen through increased positive
selection or relaxed purifying selection. For this purpose the authors could use the direction of
selection test (Stoletzki and Eyre-Walker 2011 *Mol Biol Evol* 28: 63-70) or related tests (e.g. Eyre-
Walker and Keightley 2009 *Mol Biol Evol* 26: 2097-2108). Or if polymorphism data aren't available

for these tests, simply describe the dN/dS results with reference to rates of divergence rather than
purifying selection.

**Author response:** This is a good point, but sadly the polymorphism data available to us at present
are not sufficient for performing these kind of tests. As such we have modified our description of the
dN/dS results with reference to rates of divergence rather than purifying selection as suggested.

2. Genes on autosomes versus sex chromosomes show different patterns of sex-biased
expression and different rates of evolution, and dosage compensation can contribute to sex-biased
expression on the X or obscure it. What happens to the X with the transition to asexuality? E.g. one
possible scenario is that sexual species have one copy of the X turned off, and asexual species
transition to expressing both copies; if male-biased genes are over-represented on the X, as they are
in some species, then male-biased genes will be expressed more in asexuals as a consequence of the
relaxed suppression of one X.

Are genes mapped such that the authors can test for different patterns between autosomes and X?
If not perhaps the authors could discuss whether sex chromosome dynamics might have contributed
to the observed patterns.

**Author response:** Great point! We agree dosage compensation (and its decay in asexuals) could
potentially influence the masculinization of sex-biased gene expression that we observe. We have
however looked into this in two ways and find that dosage compensation does not influence the
results we present.

Firstly, the X chromosome in *Timema* is the same in all species of *Timema* (Schwander and Crespi,
2009, unpublished results), but sex-biased genes (and in particular sex-biased genes that differ in
expression between sexuals and asexuals) are typically different between species (Fig. 5). This
means that changes to gene expression on the X (caused by altered dosage compensation) cannot
explain our results as we would then see the same genes changing in all *Timema* asexual species. We
have added this explanation to the manuscript to highlight this issue (lines 313-321).

Secondly, as part of another project we have identified the X-chromosome in each species of
*Timema*. When we repeat our analysis using only X-linked or autosomal genes shifts in sex-biased
gene expression for the X and autosomes both show the same masculinization pattern.

3. It would be more precise to say that the genome of asexuals is defeminized, and to a lesser
extent, masculinized. There's a more consistent decrease in asexuals in the bias of female-biased
genes than there is an increase in the bias of male-biased genes.

**Author response:** While we understand the reviewers point, we follow the idea that defeminization
and masculinization are synonyms since both an increase in male-biased gene expression and a
decrease in female-biased gene expression in females result in a more male-like expression pattern.
As such, for simplicity, we refer to the decreased expression of female-biased gene expression and
increased expression of male-biased as masculinization in line with previous studies (e.g. Hollis et al.
2014; Veltsos et al. 2017). We now clarify this point in the introduction (lines 57-59).

4. How were unbiased genes defined? Are they any genes that weren't male- or female-
biased? Defining them this way is likely to include many genes that are actually sex-biased but where
the magnitude is small or variable enough that they don't pass the FDR threshold. Female-biased
genes are likely to be over-represented in an unbiased gene set defined in this way, as they often
have lower magnitude of bias.

**Author response:** Yes, unbiased genes were defined as those that were not sex-biased. Whilst it is
possible that this may produce a bias towards including more female-biased genes in our unbiased
set this is unlikely to influence our results for two reasons.

Firstly, the inclusion of more female-biased genes into the unbiased set is more likely to occur in
when using a fold-change threshold than when using a significance only approach. Since we observe
qualitatively the same result with both approaches, this shows that any effect is likely small.

Secondly, inclusion of female-biased genes in the unbiased set would reduce the power to detect
differences between unbiased and female-biased genes. Since this is the strongest effect in our data
a tighter definition of unbiased genes is unlikely to change our results much. We have clarified how
unbiased genes were determined in the manuscript (lines 433-434).

5. What happens with body size with the transition to asexuality? Is there sexual size
dimorphism in sexual species, and do asexual females change towards an intermediate size, as might
be expected if they are covering more niche space that was split between the sexes in sexual
species? If this is the case then many of the metabolic and developmental process GO terms might
relate to changes in growth and body size.

**Author response:** There are no striking differences in morphology between sex and asex sister
species, with females in several pairs being difficult to distinguish from each other (now shown in
Fig. 1B; species identification of females is based on the shape of the subgenital plates and cerci).
That said, they are herbivores so niche shifts to use different parts of the plant would be expected to
have subtle effects on morphology (e.g. for instance on the mandibles as we mentioned in the
manuscript (line 309-311)), but there is not enough work in this area for us to say anything more
definitive than the speculation we already have in the manuscript on this (lines 306-312).

6. Line 108: what level of GO term? One can imagine that more fine-grained terms might vary a
great deal but higher order terms might be shared across gene sets.

**Author response:** Here the small but significant overlap in enriched GO terms for sex-biased genes
is for all levels of GO together. This approach likely produces a bias towards finding a significant
amount of overlap since enriched terms are non-independent of each other (due to hierarchical
nature of GO terms). It is however possible that the complexity of the GO term hierarchy could lead
to some functional processes being overlooked. For instance, if a GO term is enriched in one
comparison, but its parent term is enriched in another comparison, then there would be no
apparent overlap. To address this, we now also looked at the amount of 'linked overlap' of GO
terms, whereby significant GO terms were first clustered together based on parent or child terms.
Applying this approach increased the number of shared GO terms, but the amount of overlap is still
modest despite the fact that this approach is quite anti-conservative. We present this additional
analysis in the Supplemental Materials (lines 21-32) and Supplemental Fig. 2 and refer to it in the
results at lines 104-105.

7. Minor comments and typos
Headings in figure 2 – B and C are both labelled 'Male-limited' and it might help the reader grasp the
meaning more quickly if they are given different headings

**Author response:** Altered

Line 536 typo 'od'

**Author response:** Corrected

Line 546 genes should be gene

**Author response:** Corrected

The Methods describes the tissues as ‘tissue type’, which seems appropriate as whole body is not
really a tissue. The rest of the MS refers to tissues, though. It would be clearer to stick to ‘tissue
type’ throughout the MS.

**Author response:** Corrected throughout the manuscript

Line 17: Alternative splicing is another major form of resolution, so one might phrase this as e.g. that
the resolution includes genes being expressed at a different level.

**Author response:** Corrected by changing “The resolution” to “A resolution”

Line 247-248: In some systems there’s support for faster evolution of sex-biased genes through
positive selection, but in other systems, there’s support for rapid evolution through relaxed
constraint (Mank 2017 Nature Ecology and Evolution, The transcriptional architecture of phenotypic
dimorphism)

**Author response:** We have added this point and reference to the manuscript (line 247-248)

Line 365-366: I think it is worth stating in the main text that there are only whole body samples for
virgin females, to save the reader from having to refer to the supplemental materials for this.

**Author response:** We have added this to the manuscript.

Reviewer #2 (Remarks to the Author):

I very much enjoyed reading the manuscript by Parker et al and think it makes a very important
contribution to the literature. They contrast gene expression between asexual/sexual pairs of
Timema stick insects with the aim of testing whether females are constrained from evolving towards
their transcriptional optima by sexual conflict. Instead, they find that gene expression is
masculinized and attribute this to a change in fitness optima in asexual females. The approach is
novel, the manuscript is very thorough and extremely well written. It deals with a hot topic in
evolutionary biology at the moment, to what extent does sexual conflict act as a constraint and what
is the role of gene expression in its resolution, and will therefore appeal to a wide readership. I
particularly like the new perspective they provide on focusing on shifts in female fitness optima. I
have a number of comments.

The main focus of the manuscript is on sex-biased genes. The authors do mention that while
differential gene expression can resolve conflict, often the conflict is only partially resolved.
Therefore, it does makes sense to study shifts in expression of sex-biased genes when sexual conflict
is eliminated. However, I would have thought that a more interesting group of genes are those with
unbiased expression. While many unbiased genes might not be subject to conflict, theory predicts
that many are under strong sexual antagonism. But due to pleiotropy etc this conflict cannot be
resolved by differential gene expression evolution. Unbiased genes are therefore 'hotspots' of
unresolved sexual conflict. I wonder instead whether the authors could focus on these genes and
identify whether there are any convergent shifts in expression in across asexual/sexual pairs? We
might expect to see a subset of unbiased genes with consistently higher expression in asexual
females. From Fig 1.

there does appear to be shifts in expression for unbiased genes, and it would be interesting to see if
the same sets of genes show similar patterns across pairs. I wonder if this is more of a direct test of
the effect of eliminating sexual conflict and would also provide insight into the types of genes
subject to the greatest conflicts in sexual species.

**Author response:** We agree that strongly sex-biased genes likely represent mostly cases of resolved
conflict, but recent work suggests that mildly sex-biased genes are especially likely to be subject to
ongoing conflict (Cheng & Kirkpatrick 2016, Wright et al 2018). Completely unbiased genes are
unlikely to be subject to ongoing conflict given sex-biased gene expression evolves rapidly. While not
perfect, focusing on sex-biased genes at least allows us to examine a set of genes that are likely to

be enriched for sexual antagonism. Genes that change convergently across pairs are unlikely to be
linked to sexual antagonism because the genes subject to ongoing conflict are mostly different
between sexual species. Consistent with this view, we identified genes that showed convergent
shifts in expression following a transition to asexuality in previous work (Parker et al. 2019). In that
paper we found evidence for convergent gene expression changes, but little evidence that
convergent changes were due to the release of sexual conflict. The reason for this, in part, is likely
due to the fact that sexually antagonistic alleles are different in different species and ephemeral in
time as sex-specific regulation evolves.

The manuscript would benefit from a figure explicitly showing the phylogenetic relationships
between the five pairs of asexual/sexual species. Are the pairs equally distantly related? Variation in
divergence time could help interpret why the magnitude of expression shifts differ across pairs of
species. Did the authors look into this? This could provide insight into how quickly expression shifts
can evolve.

**Author response:** We have now added a figure (Fig. 1) showing the phylogenetic relationships
between the species, which also shows the relative divergence time between each of the sex-asex
pairs. We now also examine the relationship between divergence time and magnitude of expression
shifts. We found that while shifts in sex-biased gene expression changed with divergence, the effect
was both small and inconsistent between tissues and include this finding in the manuscript (Lines
116-120), Supplemental Fig. 3 and Supplemental Materials (lines 34-40), Supplemental Table 17.

A related point is whether the extent of sexual conflict in the sexual species is thought to be similar?
Do the sexual species have similar mating systems and male promiscuity?

**Author response:** We do not have a good measure of the level of sexual conflict in these species.
The mating system appears to be similar with males multiply mating in each species and with each
species showing a similar level of sexual dimorphism. We have now added photos of males and
females to Fig. 1B to show this. Of course, it would be great to know how the level of sexual conflict
varies between species and if this relates to what we observe (and hopefully in the future we will
have some good data on this!), however in this study the fact that there is sexual conflict in the
sexual species but not in the asexual species we compare them to is sufficient to test our
hypotheses.

It's great to see the authors discuss confounding effects of allometry for gene expression analyses.
Do they know whether there might be morphological differences across asexual/sexual pairs? Does
relative ovary size differ? Are there differences in egg production? I am confident that the stringent
241 fold change threshold they have imposed has likely eliminated any problems due to allometry. I also
think they can use Fig 1 to reinforce this. Median expression of unbiased genes appears not to have
changed between sexual and asexual females in each pair. If there were substantial allometry
differences I think they would detect shifts in the median expression of unbiased genes, which they
don't. It would be worth mentioning this in the manuscript to reinforce that allometry isn't a
problem here.

**Author response:** Female sexual-aseual sister species are often difficult to distinguish
morphologically (see also reply to reviewer 1). Relative ovary size and egg production rates are also
not noticeably different between sex-asex pairs. We now show images of each species used in our
study in Fig. 1B. We are glad to read that the reviewer is happy with our treatment of potential
allometry, however, we feel that a shift in in the median expression of unbiased genes is only likely
to occur when the allometry between tissues is particularly strong and thus unlikely to be useful as a
diagnostic metric for detecting allometry in most cases.

I get that the authors are trying to illustrate that shifts in expression are not specifically linked to
gene function. But unfortunately, I am not overly convinced by Fig 4. Fig 4A and B don't present
anything new in the literature, I think it is quite well established that sex-biased genes change
quickly across species. Also, the trend that masculinization is stronger for genes that are sex-biased
in sexual species seems quite noisy in Fig 4C. For example, in the legs this pattern is driven by only
one of the five data points. Instead, the authors could make this more convincing by conducting a
statistical test to assess this relationship. I am also not clear how the y axis is calculated – is this an
average across all asexual females?

**Author response:** We agree with the reviewer that it is quite well established that sex-biased genes
change quickly across species. However, we put emphasis on this point because it shows that our
finding of masculinised gene expression is not driven by gene ID but by the fact that genes are sex-
biased. This point is now even more important in our revised manuscript as it helps us to discount a
potential explanation of our results, the decay of dosage compensation, suggested by reviewer 1
(above) (lines 312-321). We have now also added a statistical test of the relationships in figure 4C

(now fig 5C in the revised manuscript) as suggested by performing a permutation ANOVA for male-
and female- biased genes in each tissue. This found that the relationships we show in Fig 4C are
statistically significant ($FDR < 0.005$) for male- and female- biased genes in each tissue type. We have
added this to methods (lines 448-452), and results (lines 190-191). We have also clarified what is
plotted on the y-axis in the legend (lines 571-572).

There is quite a strong emphasis on gene ontology analyses in the paper. The authors themselves
discuss the limitations of this approach in Timema due to unreliable gene annotation models. The
analyses are also quite descriptive. This is the weakest part of the paper and detracts from the rest
of the manuscript. I would scale down this discussion (or move it to the SI) and use the space to
explore the analyses I recommend above.

**Author response:** Although the gene ontology analyses take up quite a bit of space in the results
section, there is relatively little space given to it in the discussion itself. We are reluctant to reduce
the section on gene ontology as it is important for exploring our unexpected results from the gene
expression analyses (as praised by reviewer 1), and we feel that reducing this section would leave
the reader wondering why we did not use these approaches to try and understand our results.

The authors mention that they examine three different tissues types but really these are not
independent. Whole body is an amalgamation of reproductive tract and legs. I am not sure what the
whole-body results add to the message and would move this to the supplementary material.

**Author response:** Whilst the whole-body is not a tissue in itself, we believe it still strengthens our
findings, as though not completely independent, the whole-body is still quite different
transcriptomically to the legs and reproductive tract, as illustrated by the largely non-overlapping set
of sex-biased genes.

The species names used to refer to each asexual/sexual pair vary across figures, tables and the
manuscript (e.g. the species names in Fig 1 differ to ST5). This is a bit confusing. A figure of the
phylogeny would help with this but I would recommend referring to the same focal species
consistently.

**Author response:** We have standardized this and added a phylogeny (Fig. 1).

L20 Missing 'in'

**Author response:** Corrected

L124 I don't quite follow this sentence. There is no Y chromosome so why do the authors think that
sex-limited genes have different genetic architecture than sex-biased genes?

**Author response:** We made a mistake here. We meant sex-specific regulation rather than different
genetic architecture. We have corrected this in the manuscript.

L149 Typo – repeated were

**Author response:** Corrected

L447 Typo – was should be were

**Author response:** Corrected

L463 Typo – remove .

**Author response:** Corrected

L468 Typo – remove .

**Author response:** Corrected

L489 Typo – remove with

**Author response:** Corrected

L498 Typo – remove for

**Author response:** Corrected

The formatting of the supplementary tables varies. Sometimes column names are in bold and in
others not. Some have shaded boxes and others not.

**Author response:** We have standardized the supplemental tables

Fig 1 Legend L541 Typo – gene

**Author response:** Corrected

Fig 2 Legend L531 Typo – of

**Author response:** Corrected

Fig 2 Legend L531 ST 6 & 7 do not list the number of sex-limited genes and so I am not sure it is
relevant to refer to them here.

**Author response:** The tables do list the numbers of sex-limited genes, but this was confusing as the
column that shows them was not included in the table description. We have now clarified the table
descriptions so this information is clear.

**References**

Cheng C, Kirkpatrick M (2016). Sex-Specific Selection and Sex-Biased Gene Expression in Humans and
Flies. *PLoS Genet* 12: e1006170.

Hollis B, Houle D, Yan Z, Kawecki TJ, Keller L (2014). Evolution under monogamy feminizes gene
expression in *Drosophila melanogaster*. *Nat Commun* 5: 3482.

Parker, D. J., Bast, J., Jalvingh, K., Dumas, Z., Robinson-Rechavi, M., Schwander, T. 2019. Repeated
evolution of asexuality involves convergent gene expression changes. *Molecular Biology and*
*Evolution*. 36: 350–364

Veltsos P, Fang Y, Cossins AR, Snook RR, Ritchie MG (2017). Mating system manipulation and the
evolution of sex-biased gene expression in *Drosophila*. *Nat Commun* 8: 2072.

Wright AE, Fumagalli M, Cooney CR, Bloch NI, Vieira FG, Buechel SD, et al. (2018). Male-biased gene
expression resolves sexual conflict through the evolution of sex-specific genetic architecture. *Evol*
*Lett* 2: 52–61.

Reviewers' Comments:

Reviewer #1:

Remarks to the Author:

I found the original manuscript to be strong. The revised manuscript is nicely strengthened. Figure 1 is a great addition.

I am satisfied with the reviewers' responses to nearly all of my comments, as well as those of the other reviewer.

I have two further comments.

Response to my point 2:

The authors had two explanations for why dosage compensation was unlikely to have influenced the results. The second explanation is fully convincing, but I'm not in agreement with the first explanation: that different species have different sets of sex-biased genes (without any evidence of X-chromosome turnover that could explain different sets of genes even if dosage compensation was a factor). The authors are suggesting that if shifts in dosage compensation did explain changes in sex-biased expression, then we'd see the same gene sets popping up in each species comparison. In practice, though, the gene-by-gene tests for sex-biased expression are so underpowered (generally speaking in studies in this field) and the expression data likely to be so noisy that I would usually expect different gene sets to meet the FDR threshold even if a common process explains them. Furthermore, dosage compensation might not be regulated at the chromosome level, such that it's not clear that we should expect very overlapping sets of genes even if there was good power to detect expression differences.

The second explanation – that the results don't change whether autosomes or the X are tested – is more powerful and I wonder if the authors would consider using this in the paper instead.

Response to my point 4.

My apologies, I realized I hadn't phrased my point clearly. Another way to categorize unbiased genes is to take not only the set of genes that fails to pass FDR for sex bias, but to include a 'reversed' threshold change, i.e., to be considered unbiased a gene must show less than some magnitude of change, in addition to not being significantly differently expressed. This avoids having the unbiased gene set being 'all weakly or inconsistently sex-biased genes + truly unbiased genes'. It means leaving some ambiguous genes out of the analyses as not being clearly sex-biased or unbiased. However, I accept the authors' good point that the categorization is unlikely to influence the overall result much in this case.

Reviewer #2:

Remarks to the Author:

I am very happy with the revisions the authors have conducted and the justification for many of points I raised initially. It is a very nice paper and will make an important contribution to the field.

Author response: We would like to thank the reviewers for their constructive comments which have helped us to improve our manuscript. Our detailed responses to each point are given below.

REVIEWERS' COMMENTS:

Reviewer #1 (Remarks to the Author):

I found the original manuscript to be strong. The revised manuscript is nicely strengthened. Figure 1 is a great addition.

I am satisfied with the reviewers' responses to nearly all of my comments, as well as those of the other reviewer.

I have two further comments.

Response to my point 2:

The authors had two explanations for why dosage compensation was unlikely to have influenced the results. The second explanation is fully convincing, but I'm not in agreement with the first explanation: that different species have different sets of sex-biased genes (without any evidence of X-chromosome turnover that could explain different sets of genes even if dosage compensation was a factor). The authors are suggesting that if shifts in dosage compensation did explain changes in sex-biased expression, then we'd see the same gene sets popping up in each species comparison. In practice, though, the gene-by-gene tests for sex-biased expression are so underpowered (generally speaking in studies in this field) and the expression data likely to be so noisy that I would usually expect different gene sets to meet the FDR threshold even if a common process explains them. Furthermore, dosage compensation might not be regulated at the chromosome level, such that it's not clear that we should expect very overlapping sets of genes even if there was good power to detect expression differences.

The second explanation – that the results don't change whether autosomes or the X are tested – is more powerful and I wonder if the authors would consider using this in the paper instead.

Author response: We agree that the second explanation is more powerful, however Nature communications policy does not allow unpublished data to support inferences. Making the data we used to identify the X available is not possible at the present time as it is part of a large collaborative project. That said, what we were able to do is to use the published linkage groups for *T. cristinae* to show that the pattern of masculinization is qualitatively the same for the *T. cristinae* – *T. monikensis* species pair when X-linked transcripts were excluded, showing that masculinization is not caused by changes in dosage compensation. We have added this to the results (lines 404-408), methods (lines 554-562) and added supplementary figure 12.

Response to my point 4.

My apologies, I realized I hadn't phrased my point clearly. Another way to categorize unbiased genes is to take not only the set of genes that fails to pass FDR for sex bias, but to include a 'reversed' threshold change, i.e., to be considered unbiased a gene must show less than some magnitude of change, in addition to not being significantly differently expressed. This avoids having the unbiased gene set being 'all weakly or inconsistently sex-biased genes + truly unbiased genes'. It means leaving some ambiguous genes out of the analyses as not being clearly sex-biased or unbiased.

However, I accept the authors' good point that the categorization is unlikely to influence the

overall result much in this case.

Author response: We can see that we did mis-understand this comment previously. We agree with the reviewer that this issue is unlikely to influence the overall results, but think that leaving some ambiguous genes out of the analyses would likely be anti-conservative. We have further examined this issue by using the virgin sexual females to independently verify sex-biased genes in whole-body samples. In this way we make a set of sex-biased genes that are robust (called in two separate data sets) which we then compare to all other genes. The pattern of masculinization is qualitatively the same using this very-conservative approach. We have added this to lines 536-538 and added supplementary figure 14.

Reviewer #2 (Remarks to the Author):

I am very happy with the revisions the authors have conducted and the justification for many of points I raised initially. It is a very nice paper and will make an important contribution to the field.

Author response: We thank the reviewer for their previous comments which allowed us to improved our manuscript.